# First Descriptive Analysis of the Faecal Microbiota of Wild and Anthropized Barbary Macaques (*Macaca sylvanus*) in the Region of Bejaia, Northeast Algeria

**DOI:** 10.3390/biology11020187

**Published:** 2022-01-25

**Authors:** Mourad Boumenir, Jean-Luc Hornick, Bernard Taminiau, Georges Daube, Fany Brotcorne, Mokrane Iguer-Ouada, Nassim Moula

**Affiliations:** 1Department of Veterinary Management of Animal Resources, Faculty of Veterinary Medicine, University of Liege, 4000 Liege, Belgium; Mourad.Boumenir@doct.uliege.be (M.B.); jlhornick@uliege.be (J.-L.H.); 2Fundamental and Applied Research for Animal and Health (FARAH), University of Liege, 4000 Liege, Belgium; bernard.taminiau@uliege.be (B.T.); Georges.Daube@uliege.be (G.D.); 3Food Microbiology, Department of Food Sciences, Faculty of Veterinary Medicine, University of Liege, 4000 Liege, Belgium; 4Behaviour Biology Lab, Research Unit SPHERES, Department of Biology, Ecology, and Evolution, Faculty of Sciences, University of Liège, 4000 Liege, Belgium; fbrotcorne@uliege.be; 5Associated Laboratory in Marine Ecosystems and Aquaculture, Department of Biological Sciences of the Environment, Faculty of Nature and Life Sciences, University of Bejaia, 06000 Bejaia, Algeria; imokrane@gmail.com; 6GIGA Animal Facilities, Liege University, 4000 Liege, Belgium; 7Department of Animal Production, Faculty of Veterinary Medicine, Liege University, 4000 Liege, Belgium

**Keywords:** Barbary macaque, faecal microbiota, tourist provisioning, Gouraya national park

## Abstract

**Simple Summary:**

The gut microbiota is very important for animal physiology and health. It has been demonstrated that the gut microbiota composition of several primate species is influenced by a variety of anthropogenic factors. However, these aspects are not documented for the gut microbiota of the endangered wild Barbary macaque. This study is the first to characterize the faecal microbiota of the species and investigate the impact on it of tourist food provisioning by comparing two groups of Barbary macaques: a tourist-provisioned group and a wild-feeding group. Our results revealed the presence of 209 bacterial genera from 17 phyla in the faecal microbiota of Barbary macaques. Firmicutes was the most abundant bacterial phylum, followed by Bacteroidetes and Verrucomicrobia. The tourism activity was associated with a significant alteration of this profile, probably due to tourist provisioning issues. Increasing risks of obesity and illness call for special management measures to reduce the provisioning rate in tourist areas.

**Abstract:**

Previous research has revealed the gut microbiota profile of several primate species, as well as the impact of a variety of anthropogenic factors, such as tourist food supply, on these bacterial communities. However, there is no information on the gut microbiota of the endangered wild Barbary macaque (*Macaca sylvanus*). The present study is the first to characterize the faecal microbiota of this species, as well as to investigate the impact of tourist food provisioning on it. A total of 12 faecal samples were collected in two groups of *M. sylvanus* in the region of Bejaia in Algeria. The first group—a tourist-provisioned one—was located in the tourist area of the Gouraya National Park and the second group—a wild-feeding one—was located in the proximity of the village of Mezouara in the forest of Akfadou. After DNA extraction, the faecal microbiota composition was analysed using 16S rDNA sequencing. Statistical tests were performed to compare alpha diversity and beta diversity between the two groups. Non-metric multidimensional scaling analysis (NMDS) was applied to visualize biodiversity between groups. Behaviour monitoring was also conducted to assess the time allocated to the consumption of anthropogenic food by the tourist-provisioned group. Our results revealed the presence of 209 bacterial genera from 17 phyla in the faecal microbiota of Barbary macaques. Firmicutes was the most abundant bacterial phylum, followed by Bacteroidetes and Verrucomicrobia. On the other hand, the comparison between the faecal microbiota of the two study groups showed that tourism activity was associated with a significant change on the faecal microbiota of *M.sylvanus*, probably due to diet alteration (with 60% of feeding time allocated to the consumption of anthropogenic food). The potentially low-fibre diet at the tourist site adversely influenced the proliferation of bacterial genera found in abundance in the wild group such as *Ruminococcaceae*. Such an alteration of the faecal microbiota can have negative impacts on the health status of these animals by increasing the risk of obesity and illness and calls for special management measures to reduce the provisioning rate in tourist areas.

## 1. Introduction

The gastrointestinal microbiota, or the faecal microbiota, is defined as the set of microbial communities—viruses, bacteria, archaea, fungi, protists and their genetic material—that colonise the gastrointestinal tract of animals [1].

Prior research has shown that these microbial communities play an important role in various physiological functions of the host, including digestion and immunological defence [2,3]. Firstly, energy from food fibre is recovered exclusively by the gut microbiota. This is because mammals do not possess the enzymes—glycoside hydrolases, polysaccharide lyases and carbohydrate esterases—required to disrupt the β-1,4 glycosidic links of complex plant polysaccharides [2], the decomposition and fermentation of which allow the production of energy-rich short-chain fatty acids [4]. This function of the gut microbiota is essential for many mammalian species, including primates, which depend on plant material for a variable amount of their energy requirements [5]. Primates can obtain 30–57% of their daily energy budget from short-chain fatty acids [5,6]. Secondly, gut microbiota is also involved in the maintenance of the immune response. A disruption of gut microbiota equilibrium is associated with susceptibility to infection, decreased proliferation of intestinal lymphocytes and macrophages and low serum immunoglobulins levels (particularly IgA) [3]. Thirdly, the gut microbiota of obese individuals has been shown to have a higher metabolic potential than that of lean individuals [7]. This indicates an increased ability of this microbiota to obtain energy from the diet [7]. Furthermore, transplanting the microbiota from obese to lean individuals induced a significant increase in body fat production in these individuals [7]. This increase is linked to the proliferation of certain bacterial taxa, in particular those of the Bacterioides phylum [8]. As a result, these finding suggest that the gut microbiota have a role in the pathophysiology of obesity [7,8]. Furthermore, diabetes [9], Alzheimer’s disease [10] and Crohn’s disease [11] have all been linked to the GI microbiota dysfunction. Finally, genes specific to the microbiota can code for metabolites required by the host such as vitamin B12, biotin and folic acid [12].

There is currently a substantial body of studies on primate faecal microbiota (covering all primate families) [1]. Pioneering studies in primates attempted to investigate the bacteria colonising the fore stomach of colobines [1]. Since then, a number of studies have attempted to characterise the diversity of microbial communities associated with the gastrointestinal tract of primates, from an evolutionary, clinical and ecological perspective [1,13,14]. These studies reveal that three key drivers influence the faecal microbiota of primates: host phylogeny, transient host-related factors (sex, age, health status, behaviour, etc.) and, finally, environment (diet, seasonality, habitat quality and anthropogenic disturbance) [1]. It is now well established from a variety of studies, that diet strongly affects the microbiota composition of primate species [15,16,17,18]. One notable examination of diet–faecal microbiota relationships provided evidence that the gastrointestinal microbial composition of wild black howler monkeys vary with diet seasonal changes [15]. This finding suggests that microbial variation may help howlers meet their nutritional needs during periods when the food available in their habitat is less energetically favourable [15].

In the context of conservation, some of the anthropogenic disturbances that are causing the decline of primate populations (e.g., habitat degradation, captivity and food provisioning) were associated with faecal microbiota alterations [13,19,20]. The evidence of microbiota alteration by habitat degradation can be clearly seen in the case of red colobus (*Procolobus gordonorum*) [13]. The authors investigate the effects of habitat degradation and fragmentation on the faecal microbiota by analysing the phylogenetic and functional diversity of the gut microbiota of different groups of red colobus [13]. The results of this study showed significant differences between degraded and non-degraded habitats, with significantly higher faecal microbiota diversity in non-degraded habitats [13]. On another hand, recent evidence suggests that, in tourist areas, the intake of anthropogenic food, low in fibre and high in non-structural carbohydrates, is determinantal to the microbiota of primate species [17,20]. Chen et al., [20] demonstrated that food provisioning modifies the gut microbiota of rhesus macaques (*Macaca mulatta*). Indeed, the results of this study indicate the presence of a large variation in the richness and structure of the bacterial community between provisioned and wild rhesus macaques. Furthermore, wild macaques showed a higher abundance of bacterial communities involved in several important physiological and metabolic processes such as glycan biosynthesis and metabolism, transport and catabolism [20].

The Barbary macaque, *Macaca sylvanus*, is among the most threatened primate species in Africa (classified as an endangered species by the IUCN) [21]. This species is subject to high anthropogenic disturbances leading to a dramatic decline in wild populations [22,23]. Currently, there are few studies examining the microbial communities of the gastrointestinal tract of Barbary macaques. These limited studies have focused on specific pathogenic microbial taxa or on those of bacteria with antibiotic resistance [24,25,26]. Several known pathogenic microbes such as *Entamoeba* sp., *Iodamoeba butschlii* ([25], *Leptospira* spp., *Treponema* spp., *Mycobacterium* spp., *Acinetobacter* spp., *Rickettsia* spp. and Kinetoplastida (*Bodo* sp.) [26] have been identified in Barbary macaque’s faecal samples. In addition, previous research revealed the presence of E. coli M076 isolates resistant to colistin, b-lactams, aminoglycosides and fluoroquinolones [24].

To the best of our knowledge, both the composition of all microbial communities colonising the gastrointestinal tract of Barbary macaques, and the impact of food provisioning on these communities were not previously reported. Therefore, the goals of the present study are (1) to characterize, for the first time, the bacterial communities of the faecal microbiota of *M. Sylvanus* and (2) to explore its alteration due to tourist-food provisioning. The present study also included behavioural observations (3) to examine the diet composition of a tourist-fed group.

## 2. Materials and Methods

### 2.1. Study Sites and Groups

The samples collected in the present study were obtained from individuals of two Barbary macaque groups. The first group (tourist-provisioned group, TPG) was located in Cap Carbon (05°06′00′’ E, 36°46′00′’ N), a tourist area of the Gouraya National Park, Bejaia region, Algeria. The macaques of this group illicitly received daily food from tourists [27]. The second group (wild-feeding group, WFG) was located in the proximity of the village of Mezouara in the forest of Akfadou, Bejaia region, Algeria (4°33′16″ E, 36°17′39″ N). The absence of tourism-related disturbances, field crops (villagers were forced, several years ago, to stop farming) and no access to garbage and dumps means that this group fed almost exclusively on natural food. However, these macaques have access to the few remaining fruit trees in the site (fig and pomegranate trees).

### 2.2. Behavioural Data Collection

The TPG’s behaviour was monitored during three months from August to October 2020 using “10-min focal sampling” as observation method [28] during which feeding time, nature and provenance of consumed food items were recorded. In total, we recorded 3070 min of observation on 12 identified adult individuals (6 females and 6 males). Similar detailed observation of the WFG’s behaviour and diet composition was not possible because of the fearful nature of the macaques, which had a flight distance of more than 20 m with the observer.

### 2.3. Faecal Sample

The faecal samples were collected between 8th and 19th of October 2020. In total, 12 faecal samples (6 samples per group) were collected. In the TPG, samples were collected from two adult females and four adult males. Two samples in the WFG were obtained from two adult females. The other four WFG samples were collected from 4 adult individuals whose sex could not be identified. Approximately 2–10 g of faeces were collected from the ground within 10 min of defecation. The samples were placed in sterile 15 mL centrifuge tubes and immediately placed in a liquid nitrogen container (−196 °C). The faecal samples were kept frozen until the DNA extraction.

### 2.4. Extraction of Bacterial DNA

Global DNA extraction was performed using the GF-1 Bacterial DNA Extraction Kit (Vivantis Technologies SdnBhd, Shah Alam, Malaysia). The kit protocol was followed with the addition of a faecal liquefaction step. This step aims to suspend the bacteria in a liquid medium and isolate them from the organic material. To do this, 2 g of faeces were placed in a centrifuge tube containing 2 mL of sterile 0.9% NaCl. The mixture was homogenised and vortexed for 1 min. The mixture was then centrifuged at 1000× *g* for 3 min. The supernatant (1–2 mL) was collected and placed in a sterile 2 mL centrifuge tube and centrifuged at 6000× *g* for 2 min. For the next step, the kit protocol was rigorously respected.

### 2.5. PCR Amplification and Product Quantification

PCR amplification of the V1-V3 hypervariable region of the 16S rDNA and library preparation were performed with the following primers (already linked to Illumina-adapters), forward (5′-GAGAGTTTGATYMTGGGCTCAG-3′) and reverse (5′-ACCGCGGCTGCTGGCAC-3′). PCR products were purified with the “AgencourtAM Pure XP beads kit” (Beckman Coulter; Pasadena, CA, USA) and subjected to a second round of PCR for indexing, using Nextera XT index primers 1 and 2. After purification, PCR products were quantified using Quant-IT PicoGreen (ThermoFisher Scientific; Waltham, MA, USA) and diluted to 10 ng/μL. Final qPCR quantification of each library sample was performed using the KAPA SYBR^®^ FAST qPCR kit (KapaBiosystems; Wilmington, MA, USA) prior to standardisation, pooling and sequencing on a MiSeq sequencer using V3 reagents (Illumina; San Diego, CA, USA). A positive control using DNA from 20 defined bacterial species and a negative control (from the PCR step) were included in the sequencing.

### 2.6. Bioinformatics Analyses

Raw sequences were processed using MOTHUR v1.41 (https://www.mothur.org, accessed on 24 October 2021) for alignment and clustering, and the VSEARCH algorithm for chimera detection [29,30,31]. As a guideline, standard MOTHUR MiSeq SOP was used to perform the reads processing and OTU generation [32]. A clustering distance of 0.03 was used for OTU generation. 16S rDNA reference alignment and taxonomic assignment were based on the SILVA (v1.38) database (https://www.arb-silva.de, accessed on 24 October 2021) of full-length 16S rDNA sequences (Quast et al., 2013). From 2,616,654 raw sequences, we retained 2,525,946 sequences after cleaning (length and sequence quality) and 2,312,869 sequences with a median length of 493 nucleotides after searching and removing chimeric sequences. A rarefied table with 10,000 sequences per sample was used for taxonomic assignment and OTU clustering. Good’s coverage estimator was used as a measure of sampling effort for each sample, with an average value of 99.77% (Appendix A). Negative controls, as a measure of determining erroneous results due to contamination, were not sequenced as there was no detectable amplification product in the samples.

### 2.7. Data Analysis

A statistical analysis of the overall scores using a paired *t*-test was used. High throughput sequencing (NGS) was performed and used to assess alpha diversity (richness estimation—Chao1 estimator, microbial biodiversity—Simpson reciprocal index and population regularity, or equitability—derived from the Simpson index) using MOTHUR software. Beta diversity (bacterial community composition) was assessed with MOTHUR using Bray–Curtis dissimilarity matrix and Jaccard compositional matrix. The variance of the microbial profiles for the two groups was compared with an analysis of molecular variance (AMOVA) test (with 10,000 permutations) [33]. Beta-dispersion was assessed with the homogeneity of molecular variance (HOMOVA) test for homogeneity of variance between the two groups (10,000 permutations) in MOTHUR [34,35]. Non-metric multidimensional scaling analysis (NMDS), based on the Bray–Curtis dissimilarity matrix, was applied to visualise biodiversity between groups [36]. Ordination analysis and 3D graphics were performed with Vegan (https://CRAN.R-project.org/package=vegan, accessed on 24 October 2021), Vegan3d (https://CRAN.R-project.org/package=vegan3d, accessed on 24 October 2021) and rgl packages (https://CRAN.R-project.org/package=rgl, accessed on 24 October 2021) in R (R: A Language and Environment for Statistical Computing, R Foundation for Statistical Computing, Vienna, Austria, 2015; https://www.R-project.org/, accessed on 24 October 2021).

Differential abundances of bacterial population between groups were assessed with DESeq2, using Deseq2 package in R [37].

## 3. Results

### 3.1. Diet Composition of the Tourist-Provisioned Group

The analysis of the TPG diet indicated that the feeding time of anthropogenic resources was dominant during the study period (61.69% of the feeding time). Sweet foods (cakes, wafers, biscuits, chocolate and sweets) were the most abundant in the diet of this group (30% of the feeding time; SD = 6.85). The proportion of pasta was also high (13.8% of the feeding time; SD = 3.72). The part of fruit, both from natural and from human origin, amounted to 17.9% of the total feeding time (SD = 3.05). The lowest feeding time proportions of anthropogenic foods in the diet were peanuts (5.7%; SD = 3.09), dairy products (1.82%; SD = 1.11) and drinks (1.5%; SD = 0.807). The consumption of leaves and seeds from the natural environment was lower, respectively, 13.6% (SD = 3.55) and 10.3% (SD = 3.24). Flowers, insects, bark and stems were the items that were the least represented in the diet of the TPG (˂1% for each one of them) (Figure 1). In some cases, macaques have been observed feeding on charcoal left behind by tourists after lighting a fire.

### 3.2. Characterisation of the Faecal Microbiota of M. sylvanus

The study of the faecal microbiota revealed the presence of 209 bacterial genera from 17 phyla. The most abundant bacterial phyla were Firmicutes (94.475%; SD: 3.514), Bacteroidetes (2.655%; SD: 2.871) and Verrucomicrobia (1.084%; SD: 1.910). The phylum WPS-2 was also found but at very low abundances (0.0603%; SD: 0.068) in three individuals from the Gouraya National Park group (Figure 2; Appendix A). 

The most abundant bacterial genera were *Ruminococcaceae*_UCG-002 (18.12%; SD: 17.04), *Ruminococcaceae*_ge (12.36%; SD: 14.28), *Ruminococcus*_2 (10.86%; SD: 13.18) and *Faecalibacterium* (8.21%; SD: 12.49). A total of 1731 operational taxonomic units (OTUs) were identified. *Erysipelotrichaceae*_UCG-003_Otu16664 was the most abundant OUT followed by *Ruminococcaceae*_NK4A214_group_Otu13512, *Ruminococcaceae*_NK4A214_group_Otu14072 and *Ruminococcus*_1_Otu00631 (Figure 3; Appendix A).

### 3.3. Evaluation of the Impact of Food Provisioning on the Faecal Microbiota of Barbary Macaques

Alpha diversity analysis revealed no significant differences for the following indices: Simpson reciprocal biodiversity index, Chao1 richness estimator for bacterial genera and the population regularity index (Figure 4). The alpha diversity of the wild-feeding group and the tourist-provisioned group were similar despite the difference observed graphically.

The NMDS analysis showed the completely distinct compositions of the faecal microbiota of the two groups of *M. sylvanus.* The points belonging to the wild-feeding group did not overlap with those of the tourist-provisioned group (Figure 5). The analysis of molecular variance (AMOVA) showed a significant difference between the wild and the tourist-provisioned group (F score: 9.37; *p*-value: 0.0017). However, the analysis of the homogeneity of molecular variance (HOMOVA) showed no significant difference between the two groups.

The proportions of bacterial genera differed significantly between the two groups (Figure 6; Appendix A). The genera *Ruminococcaceae*_UCG-002 (WFG = 28%, TPG = 8%; *p*-value = 0.04), *Ruminococcaceae*_ge (WFG = 20%, TPG = 4%; *p*-value = 0.05), *Ruminococcaceae*_UCG-005 (WFG = 12%, TPG = 4%; *p*-value = 0.0001) and *Ruminococcaceae*_UCG-010 (WFG = 7%, TPG = 0.8%; *p*-value = 0.03) were significantly more abundant in the wild-feeding group compared to the tourist-provisioned group. On the other hand, the TPG showed more abundant proportions of the genera *Ruminococcus*_2 (WFG = 06%, TPG = 21%; *p*-value = 0.007), *Faecalibacterium* (WFG = 0.5%, TPG = 15%; *p*-value = 0.04), *Christensenellaceae*_R-7_group (WFG = 1%, TPG = 6%; *p*-value = 0.006), *Subdoligranulum* (WFG = 1%, TPG = 6%; *p*-value = 0.006) and *Dorea* (WFG = 0.2%, TPG = 4%; *p*-value = 0.008). The dominant genus in the WFG was *Ruminococcaceae*_UCG-002 while the genus *Ruminococcus*_2 was the most abundant in the tourist-provisioned group.

## 4. Discussion

The first aim of the present study was to explore for the first time the faecal microbiota of Barbary macaques. The second aim was to investigate the impact of human food provisioning on this bacterial communities in a touristic site—National Park of Gouraya (Bejaia region, north-eastern Algeria). Due to the impossibility to perform a strict case-control study, a comparison was made between the faecal microbiota of a tourist-provisioned group and that of a wild-feeding group. Finally, the third aim was to determine the time allocated to the consumption of anthropogenic versus natural foods in TPG. Overall, our results showed that tourism activity significantly impacted the faecal microbiota of the Barbary macaques via a major contribution of anthropogenic food in the diet of TPG. The potentially low-fibre diet at the tourist site was not advantageous to the proliferation of bacterial genera found in abundance in the wild-feeding group such as *Ruminococcaceae*. Such a modification of the faecal microbiota could be the cause of several diseases in the tourist-provisioned group. Bauer and collaborators [35] reported the impact of provisioning on the development of obesity and type 2 diabetes in rhesus and long-tailed macaques (*M. mulatta* and *M. fascicularis*).

### 4.1. Diet Composition of the Tourist-Provisioned Group

As expected, the diet analysis of the TPG revealed that anthropogenic food was the main food resource for these macaques during the study period, as it represented 60 percent of the total feeding time. The anthropogenic diet, which consists primarily of sweet foods, pasta, dairy products and drinks (low in fibre and high in fats and non-structural carbohydrates) that can be easily digested by omnivorous primates. However, the macaques were also supplied with items, such as fruits and nuts, that may have similar nutritional quality when compared to natural food items. These could partially compensate for the reduced intake of natural resources. To confirm these hypotheses, a study of the nutritional quality of the food eaten by the macaques in Gouraya National Park is needed.

During the present study, Barbary macaques were observed ingesting tree bark and charcoal. Several authors have suggested that this unusual behaviour may be caused by mineral deficiency in the diet [38,39]. On the other hand, charcoal consumption could also be used to relieve illness as it is known to absorb certain toxins [40,41]. This behaviour could thus be another example of self-medication in primates.

Finally, it was reported previously that artificial provisioning leads to an increased dependence of generalist primates on anthropogenic resources, and a possible loss of the efficiency and acquired foraging capacity of natural resources [42].

### 4.2. Characterisation of M. sylvanus Faecal Microbiota

To our knowledge, the present study provided the first report of the phylogenetic diversity of the faecal bacterial community of the Barbary macaques. The result of this analysis showed that the dominant bacterial phyla in faecal samples included Firmicutes, Bacteroidetes, Verrucomicrobia and Tenericutes, a similar composition to that reported for other primate species [20,43,44]. However, the proportion of Firmicutes found in the present study was surprisingly higher compared to other primate species [17,20]. Clayton et al. [17] and Chen et al. [20] reported that an increased proportion of Firmicutes could improve digestion and dietary energy absorption. In these bacterial phyla, members of the *Ruminococcaceae*, *Lachnospiraceae*, Clostridiales_Family_XIII, *Christensenellaceae* and *Akkermansiaceae* families were the most abundant. These bacteria can degrade cellulose, hemicellulose and pectin, among other structural carbohydrates, and are present in the gastrointestinal tracts of numerous species [1,4,12]. The *Christensenellaceae* family plays an ecological role in the gut of animals by fermenting glucose to acetate and butyrate under anaerobic conditions, implying that it ferments sugars in the gut to short chain fatty acids and other fermentation products such as H_2_ and CO_2_ [45]. High abundance of these bacteria was associated with diarrhoea in captive Sichuan golden snub-nosed monkeys (*Rhinopithecus roxellana*) [46].

### 4.3. Impact of Food Provisioning on the Gut Microbiota of the Barbary Macaque

The gut microbiota of primates is influenced by a number of factors including genetic factors of the host, and environmental changes driving diet variations [1,17,47]. It is currently known that anthropogenic disturbance such as habitat degradation [13], food provisioning by tourist [17,20] and captivity [16,48] modulate the composition of primate microbial communities. Evidence suggests that disturbances implicating change in diet are among the most important factors inducing variations in the composition of the gut microbiota [15,17].

Counterintuitively, due to expected more monotonous food items, alpha diversity, genus richness and genus regularity of the microbial communities were found to be numerically higher in the TPG. However, our study did not find any significant difference between the two groups for these parameters. A possible explanation for this might be that the TPG still consumed natural foods [20]. Indeed, our results of the diet study indicated that, although the macaques in Gouraya National Park regularly received human food, they also drew their diet from natural resources such as leaves, seeds and fruits. On the other hand, provisioned foods included high-fibre items such as fruits and nuts. As a result, the balance between provisioned and natural food may maintain the previous diversity parameters and the ability of Barbary macaques to digest a variety of fibre-rich natural foods. Another possible explanation for this lack of significant difference is that faecal sampling was carried out during the dry season, which is marked by a major reduction in natural food resource availability [15,49]. The animals may switch to a more fibrous diet during this feed restriction phase potentially conducting to a decrease of diet diversity for the wild-feeding group. This could also explain the high abundance of bacterial species that degrade fibre food in this group. However, further research over a longer period of time is required to confirm the impact of seasonality on the GI microbiota. These results are in line with those of recent studies in rhesus macaques which also showed no differences in diversity between a provisioned and a wild group [20,50]. However, contrary to our finding, a higher bacterial community richness was found in the provisioned group [20]. On the other hand, another study reports the decrease in microbial diversity as a function of lifestyle and habitat disturbance, especially for provisioned groups, in *Pygathrix nemaeus* [17].

One interesting finding was the presence of significant differences between the two groups in the beta diversity measurements. Indeed, the faecal microbiota compositions of the two *M. sylvanus* groups were quite separated. Bray–Curtis distance analysis and Jaccard index—explored to assess the distance between the two groups of macaques—indicated that the abundances of the microbial communities were significantly different. The analysis of molecular variance (AMOVA) confirms these results which are consistent with studies in other macaque species [17,20,50].

Another important finding was that the abundances of bacterial genera vary significantly between the two groups. The incorporation of anthropogenic foods and the reduction of natural foods in the diet could be a major driver leading to a decrease in the abundances of bacterial taxa known to degrade high-fibre natural foods and an increase in the abundances of taxa known to degrade fats and sugars. Indeed, some bacterial genera were present in the WFG at significantly higher proportions than in the TPG, probably due to differences in the dietary composition of the two groups. These included four genera of the *Ruminococcaceae* family that occur as fibrolytic bacterial communities playing an important role in cellulose degradation [8,51,52]. However, the genus *Ruminococcus*_2, which is important for cellulose and hemicellulose fermentation [51,52], was more abundant in the TPG. The faecal microbiota of this group also showed a significantly higher abundance of the genus *Faecalibacterium*. This genus, represented mainly by the species *Faecalibacterium prausnitzii*, is involved in several functions in the host [53]. Previous studies considered these commensal bacteria to be bio-indicators of human health and are the taxon which produces the most butyrate in the colon and they also play an important role in the inflammatory, immunological and digestive processes [53,54]. It is interesting to note that some genera of the *Ruminococcaceae* family were associated with slim and medium phenotypes [8]. In contrast, increased abundance of the genus *Faecalibacterium* was associated with obese phenotypes [8]. Another important result found in this study was the high abundance of the genus *Akkermansia* found in the TPG. It was been demonstrated that the abundance of these bacteria increases significantly when the host diet is high in fat, as it is with TPG in this study [55]. *Akkermansia muciniphila* was recognized for its role in mucin degradation, control of host mucus turnover and has been suggested to play protective roles in the intestinal integrity [56,57]. However, an overabundance of this bacterium may cause tissue inflammation by weakening the colon’s mucus barrier [55,58]. Indeed, the use of mucin as a nutrient by an overpopulation of *A. muciniphila* can lead to thinning of the mucus layer. As a result, the intestinal barrier function is weakened and the transepithelial migration of microbes and their metabolites is facilitated [55,58].

Finally, in addition to a possible increased dependence on anthropogenic resources and loss of the acquired foraging capacity on natural resources [41], it was demonstrated previously that provisioning reduces progressively the ability of these primates to survive in their natural habitats by altering their gut microbiota’s ability to ferment foliage [59].

## 5. Conclusions

To conclude, the present study shows that tourism activity was associated with significant differences in the faecal microbiota of a tourist-provisioned group of Barbary macaques when compared to a wild-feeding group. The consumption of anthropogenic foods—potentially poor in fibre and rich in simple sugars and fats—may have modulated the genera abundances and beta diversity of the faecal microbiota in the tourist-provisioned group with negative repercussions on the health status. As a result, it is recommended for the Algerian authorities to implement special management measures to reduce the food-provisioning rate in the touristic areas.

## Figures and Tables

**Figure 1 biology-11-00187-f001:**
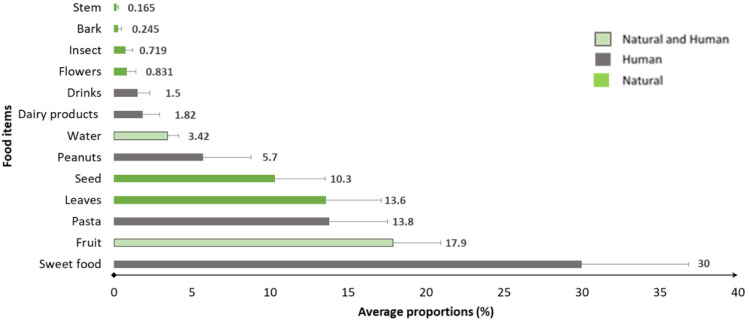
Average proportions of the different food items consumed by Barbary macaques of a tourist-provisioned group in the Gouraya National Park, Cap Carbon. Colours represent the origin of food items: green = natural food; grey = human food; green framed with grey = both natural and human food.

**Figure 2 biology-11-00187-f002:**
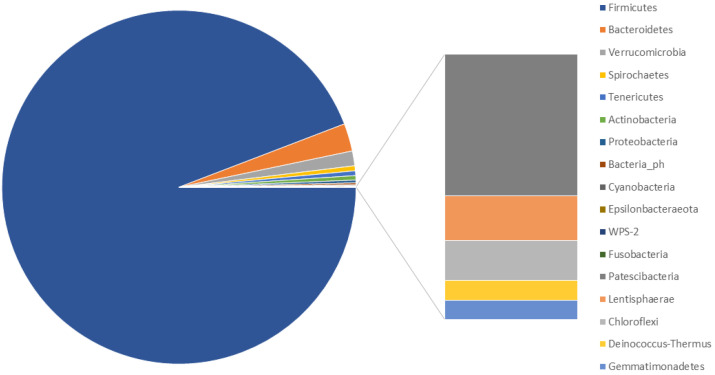
The average proportion of bacterial phyla in the faecal microbiota of *Macaca sylvanus* in the Gouraya National Park and in the Akfadou forest massif.

**Figure 3 biology-11-00187-f003:**
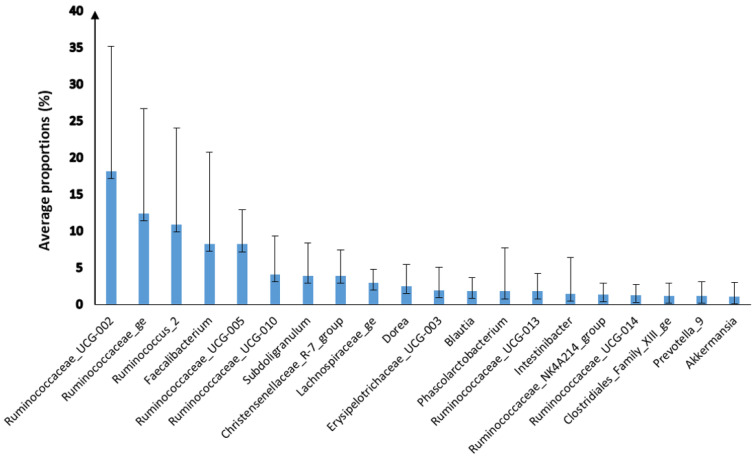
The average proportion of the most abundant bacterial genera (≥1%) in the faecal microbiota of *Macaca sylvanus* in the Gouraya National Park and in the Akfadou forest massif (SD).

**Figure 4 biology-11-00187-f004:**
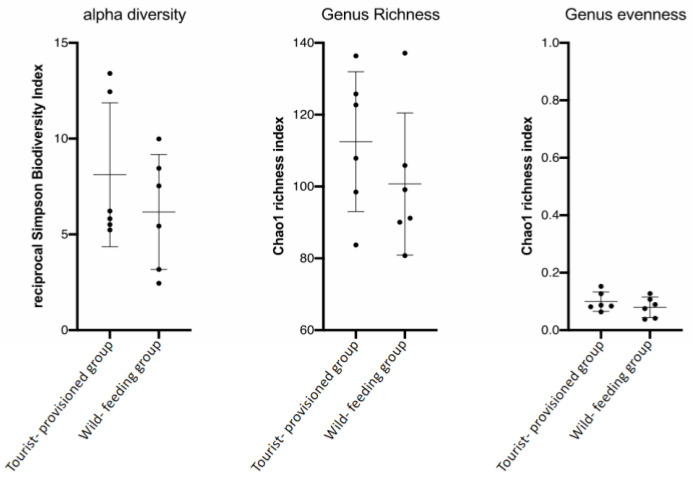
Alpha diversity, richness and regularity of bacterial genera in the faecal microbiota of two groups of *Macaca sylvanus* (a tourist-provisioned group (TPG) in the Gouraya National Park and a wild-feeding group (WFG) in the Akfadou forest massif). Data are scatter dot plots at the genus level for individual macaques in the two defined groups with the mean and standard deviation.

**Figure 5 biology-11-00187-f005:**
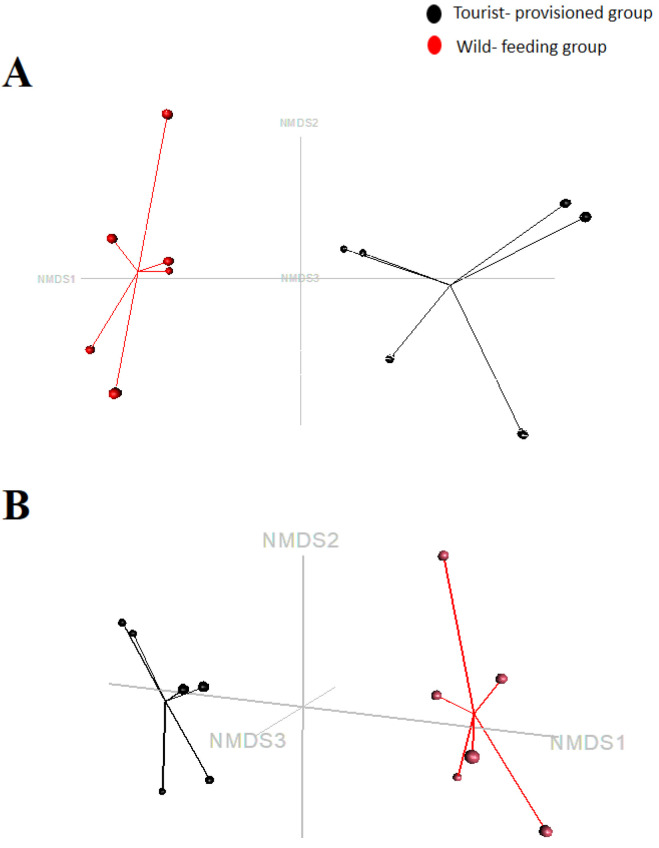
Beta diversity of the bacterial community of the faecal microbiota of two groups of *Macaca sylvanus* (a tourist-provisioned group in the Gouraya National Park and a wild-feeding group in the Akfadou forest massif) using multidimensional non-metric scaling (NMDS) plots generated by Bray–Curtis dissimilarity distances (model stress = 0.001) (**A**) and JACCARD matrix (model stress = 0.013) (**B**).

**Figure 6 biology-11-00187-f006:**
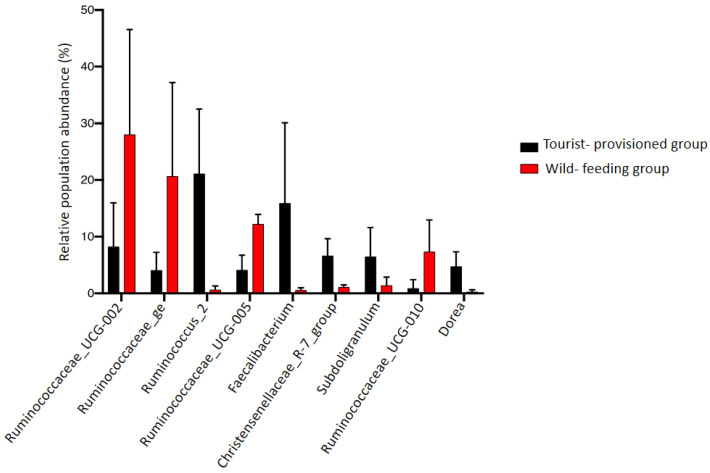
Proportions of bacterial genera in the faecal microbiota of two groups of *Macaca sylvanus* (a tourist-provisioned group in the Gouraya National Park and a wild-feeding group in the Akfadou forest massif).

## Data Availability

The data presented in this study are available from the corresponding author on request.

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
