# Peer review of "First Descriptive Analysis of the Faecal Microbiota of Wild and Anthropized Barbary Macaques (Macaca sylvanus) in the Region of Bejaia, Northeast Algeria"

_biology, 2022, doi:10.3390/biology11020187_

Round 1

Reviewer 1 Report

The sample size of this study is extremely small (N = 12 fecal samples) and thus there is unlikely to be enough statistical power for the authors to test for an impact of human provisioning on the gut microbiome of barbary macaques. This may explain the discrepancies between the visualization of the data and the results of statistical tests as described in the results section, which will heavily rely on the leverage and influence of each data point. There is a lot of detail missing in the methods section regarding the bioinformatics approaches, impeding reproducibility. The authors do not present numerical values or tables on bacterial relative abundance in any of the results section and there is no supplementary material, both of which are highly atypical for a microbiome paper. Together, these shortcomings do not give me confidence in the suitability of this paper for publication at this stage. My suggestions for the authors would be 1) to consider newer, more robust statistical approaches to analyzing microbiome data (e.g., see recent papers on linear modeling), and 2) greatly increase attention to detail in both the methods and results section by describing the bioinformatic and analytical approaches with enough nuance that they could be repeatably and by including tables with actual values for diversity and relative abundance in the results section. Short of including more samples into the dataset, I would suggest reframing the paper as a descriptive study rather than a test of the effects of provisioning on the gut microbiome for resubmission.

Author Response

Thank you for reviewing our manuscript “First analysis of the faecal microbiota of the Barbary macaque (Macaca sylvanus) reveals differences between a wild group and a tourist-provisioned group in the region of Bejaia, North East of Algeria”.

We will answer point-by-point to your comments (in italic). We hope you will be satisfied. All changes were carried out in the new version of the manuscript and track changes is activated for visibility. 

Best regards,

Reviewer 1 question 1:

The sample size of this study is extremely small (N= 12 fecal sample) and thus there is unlikely to be enough statistically power for the authors to test the impact of impact of human provisioning on the fecal microbiota of Barbary macaques. This may explain the discrepancies between the visualization of the data and the result of statistical tests as described in the result section, which will heavily rely on the leverage and influence on each data point.

Answer:

Thank you for the comment. The power depends on the statistical effect sought and also on the number of subjects included in the trial. However, a small number of subjects does not question the probability to detect significant effects. It is likely to hide real effects (beta risk), but not to reveal false positive effects (alpha risk). In our study, highly significant differences were found, what was considered as informative, to our knowledge.

Studies on primate microbiota are not often high in number, and highly diverse, fostering dispersion of mean results. So any study helps to enhance the overall sample size and therefore the power of meta-analysis, contributing to focus on real effects or to detect possible publication bias effect.

Reviewer 1 question 2:

There is a lot of detail missing in the methods section regarding the bioinformatics approaches, impeding reproducibility.

Answer:

Additional details and links to articles that further describe the methodology used in this study have been added to the section on methodologies.

Reviewer 1 question 3:

The authors do not present numerical values or tables on bacterial relative abundance in any of the results section and there is no supplementary material, both of which are highly atypical for a microbiome paper.

Answer:

The average proportions of diet composition, genera abundance and phyla abundance have been added to the results section. A supplementary material containing the following tables was added:

- Table S1 A measure of sampling effort for each sample using Good’s coverage estimator (rarefied table at 10,000 sequences per sample).

- Table S2: The relative abundance and the average proportions of the bacterial phyla of Barbary macaque fecal microbiota in both TPG and WFG.

- Table S3: The average proportions of Barbary macaque fecal microbiota at phylum level between the TP and WF groups.

- Table S4: The average proportions of Barbary macaque fecal microbiota at genus level between the TP and WF groups.

Reviewer 1 question 4:

Short of including more samples into the dataset, I would suggest reframing the paper as a descriptive study rather than a test of effects of provisioning on the gut microbiome for resubmission.

Answer:

As argued above, the sample size could be considered as sufficient because statistically significant effects were found in the statistical analyses. Moreover, the statistical analyses used have been explained in the material and methods section in the new corrected version. We also added the P values obtained in the Results section.

Following your remark we propose this new title for the article:  “First descriptive analysis of the faecal microbiota of wild and anthropized Barbary macaques (Macaca sylvanus) in the region of Bejaia, northeast Algeria.”

Reviewer 2 Report

This paper examines the gut microbiota of wild Barbary macaques and the potential influence of tourist-provisioned diets on microbiota diversity, composition, and membership. The study takes advantage of a ‘natural’ experiment whereby the diets of two macaque groups differ drastically due anthropogenic influence. The premise and results of the paper are interesting and warrant publication. However, there are a few concerns that need to be addressed prior to publication. Below, I have outlined major concerns as well more minor comments throughout.

Major concerns:

  1. In the Introduction, the authors provide background information on the general importance and function of gut microbiota, but there is a lack of more specific background on study’s specific questions and its relevance to a greater understanding of animal gut microbiota. Currently, the Introduction includes a number of very specific, yet irrelevant, details about the gut microbiota that have little to do with the study (e.g., Lines 72-75; 81-83; 90-91; 116-117). These details can be pared down to make room for more relevant information. There is a substantial body of literature on the influence of anthropogenic activities on wildlife gut microbiota and it should be included in the Introduction. I suggest the authors reframe their Introduction to be less about the gut microbiota in general and focus more on the literature that is relevant to their three aims.
  2. Regarding the Methods, it is great to see the authors employ a number of measures of alpha diversity as they can reflect different patterns in the microbial data. I suggest the authors do the same for the beta diversity data, because the different patterns reflected in different metric can be revelatory. It is generally common practice to analyze both an ‘abundance-weighted’ measure (Bray-Curtis, as is already reported) as well as a ‘presence-absence measure’ (Jaccard or unweighted Unifrac). Particularly for these data, where the presence/absence of less abundant taxa may be driving variation in the primate gut microbiota, I would be interested to see analyses of Jaccard beta diversity distance.
  3. The statistics need a greater explanation in the methods and results. In the Methods, please define the acronyms AMOVA and HOMOVA. Many would likely mistake AMOVA for ANOVA and it is important to note the distinctions. In the results, all statistical results, the name of the test being run should be included in the parentheses with the p-value, otherwise, it is unclear how the statistics were generated.
  4. In the Results for the fecal microbiota, the lists of abundant phyla and genera (Lines 209-218) are not particularly revelatory because the proportions are not reported nor represented in a figure. If the authors choose to keep the list, the average proportion and standard deviation needs to be reported for each phylum and genus. I also encourage the authors to include an additional figure (e.g., stacked bar chart or pie chart) showing the proportions of phyla and genera in the fecal microbiomes of both macaque groups, averaged across individuals. Generally, only the abundant taxa (those representing e.g., <1% of the microbiota) are included in these figures. This would allow the reader to better visualize the microbiota membership. Also, it is unclear from the methods how genera and OTU differ. Were OTUs identified down to strain level? As a general note, I encourage the authors to look into bioinformatic pipelines that generate ASVs.
  5. When comparing the abundances of specific genera between the two macaque groups, did the authors use differential abundance analyses? If not, I highly recommend the authors look into using these methods as simple comparisons of relative abundances are not robust and are subject to issues of compositionality (check out this paper for more information: https://doi.org/10.1038/s41522-020-00160-w).
  6. In the Discussion: There is quite a lot of information that repeating the results (Line 273-274; 292-293; 294-296). This should be avoided wherever possible. Although the discussion does include some interesting links to other studies, it needs to be streamlined and better referenced (see below for places where citations are needed). I also encourage the authors to be careful about assigning causality to the different diets of the two groups. Yes, there were significant differences in the microbiota of the two groups, and they can infer that diet could be a major driver. But the authors should be careful when, for example, reporting that “the reduction of natural foods in the diet led to a decrease in the abundances of bacterial taxa known to degrade high-fibre natural foods” (lines 342-343). These data are correlative.
  7. As a general note, the writing would benefit from a thorough proof-reading for grammar and overall flow. Currently, there are numerous places where the language does not work or is confusing. If English is not a first language and the author’s need assistance, I encourage them to reach out to the journal editor.

Minor comments:

General:

- References needed for sentences in Lines: 61, 64-65, 88-89, 91-92, 93-94, 296-297, 354-355

- In the results and discussion, the authors switch between past and present tense. Please pick one and standardize throughout.

- Whenever percentages or proportions are reported (for behavioral data and microbial data), if they are averages across multiple individuals, they need to also include plus/minus standard deviations (same with Figure 1, should include error bars).

- Figures are very small and difficult to read. This could be an artifact of the initial submission but for final submission, the figures should be much larger and higher resolution.

- In discussion, there are places where it is unclear whether the results being discussed are from this study or previous studies (e.g., 352-353). Be sure to clarify.

Specific:

Graphical abstract: change “genus” to “genera”

72-75: There is a lot of debate about the proportion of bacterial to hosts cells/DNA, so be careful when reporting them. Here, they are not necessary for this study and I would remove them (see comment above about Introduction).

 166-167: Were sequences from chloroplast and mitochondria filtered out? Not always necessary but should be reported either way.

169: What were the min and max sampling depths for the samples? Need to include a rarefaction plot at a supplementary figure to determine if 10,000 is an appropriate rarefying depth.

192: Please define “sweet food”. It is very ambiguous and could include fruit.

362-363: Akkermansia is no longer a monotypic genus.

Figure 1: The percentages add up to more than 100, please revise. Also, “percentages” is spelled incorrectly on the x-axis title.

Author Response

Thank you for reviewing our manuscript “First analysis of the faecal microbiota of the Barbary macaque (Macaca sylvanus) reveals differences between a wild group and a tourist-provisioned group in the region of Bejaia, North East of Algeria”.

Best regards,

We will answer point-by-point to your comments (in italic). We hope you will be satisfied. All changes were carried out in the new version of the manuscript and track changes is activated for visibility.

  1. Major concerns

Reviewer 2 question 1:

In the introduction, the authors provide background information on the general importance and the function of gut microbiota, but there is a lack of more specific background on study’s specific questions and its relevance to greater understanding of animal gut microbiota. Currently, the introduction includes a number of very specific, yet irrelevant, details about the gut microbiota that have little to do with the study (e.g., Lines 72-75; 81-83; 90-91; 116-117). These details can be pared down to make room for relevant information. There is a substantial body of literature on the influence of anthropogenic activities on wildlife gut microbiota and it should be included in the introduction. I suggest the authors reframe their introduction to be less about the gut microbiota in general and focus more on the literature that is relevant to their three aims.

Answer:

We are very grateful to the reviewer for his careful reading of the manuscript. We reframed the introduction part in the manuscript according to these recommendations. Some details on the general importance and function of gut microbiota were pared down. You can now find in the introduction section details on the drivers that influence the faecal microbiota of primates (Lines 86- 101) and some literature on the anthropogenic disturbances on primate gut microbiota (Lines 102- 119).

Reviewer 2 question 2:

Regarding the methods, it is great to see the authors employ a number of measures of alpha diversity as they can reflect different patterns in the microbiota data. I suggest the authors do the same for beta diversity data, because the different patterns reflected in different metric can be revelatory. It is generally common practice to analyze both an ‘abundance-weighted’ measure (Bray-Curtis, as is already reported) as well as a ‘presence-absence measure’ (Jaccard or unweighted unifrac). Particularly for these data, where the presence/ absence of less abundant taxa may be driving variation in the primate gut microbiota, I would be interested to see analyses of Jaccard beta diversity distance.

Answer:

We performed the suggested compositional analysis using the Jaccard (presence-absence) index. We also performed an NMDs model on it and an AMOVA test.

Reviewer 2 question 3:

The statistics need a greater explanation in the methods and results. In the methods, please define the acronyms AMOVA and HOMOVA. Many would likely mistake AMOVA for ANOVA and it is important to note the distinctions. In the results, all statistical results, the name of test being run should be included in parentheses with the p- value, otherwise, it is unclear how the statistics where generated.

Answer:

We have now corrected this omission; the definition of AMOVA and HOMOVA were added in the manuscript (Lines 211- 215). Also, the p- value was included in parentheses in the result section.

Reviewer 2 question 4:

In the results for fecal microbiota, the lists of abundant phyla and genera (Lines 200-218) are not particularly revelatory because the proportions are not reported nor represented in a figure. If the authors choose to keeps the list, the average proportion and standard deviation needs to be reported for each phylum and genus. I also encourage the authors to include an additional figure (e.g., stacked bar chart or pie chart) showing the proportions of phyla and genera in the fecal microbiomes of both macaques groups, averaged across individuals. Generally, only the abundant taxa (those representing e.g., ˂1% of the microbiota) are included in these figures. This would allow the reader to better visualize the microbiota membership.

Answer:

As you recommended, the average proportion and standard deviation for each phylum and genus were included in the results section. We also added additional figure of the average proportion of bacterial phyla (pie chart) (Lines 248- 250) and the average proportion of the most abundant bacterial genera (≥ 1%) (bar chart) (Line 257- 259) in the fecal microbiota of Macaca sylvanus.

Reviewer 2 question 5:

Also, it is unclear from the methods how genera and OTU differ. Were OTUs identified down to strain level? As a general note, I encourage the authors to look into bioinformatics pipelines that generate ASVs.

Answer:

In our viewpoint, it is not a good idea because ASV is not suitable for not well-known biotope. Furthermore, it is not relevant when you group your OTU/ASV into phylotype.

Reviewer 2 question 6:

When comparing the abundances of specific genera between the two macaque groups, did the authors use differential abundance analyses? If not, I highly recommend the authors look into using these methods as simple comparisons of relative abundances are not robust and are subject to issues of compositionality (check out this paper for more information: https://doi.org/10.1038/s41522-020-00160-w).

Answer:

We fully agree with this reviewer and additional analyses of differential abundances of bacterial population between groups were assessed with DESeq2, using Deseq2 package in R.

Reviewer 2 question 7:

In the discussion: there is quite a lot of information that repeating the results (Line 273-274; 292-293; 294-296). This should be avoided wherever possible. Although the discussion does include some interesting links to other studies, it needs to be streamlined and better referenced (see below for places where citations are needed). I also encourage the authors to be careful about assigning causality to the different diets of the two groups, and they can infer that diet could be a major driver. But authors should be careful when, for example, reporting that the reduction of natural foods in the diet led to a decreases in abundances of bacterial taxa known to degrade high-fibre natural foods “ (lines 342-343)! These data are correlative.

Answer:

On this issue, we fully agree with the reviewer. The sentence has been rewritten in order to make causality appear plausible rather than verified (Lines 393- 396).

Reviewer 2 question 8:

As a general note, the writing would benefit from a thourough proof-reading for grammar and overall flow. Currently, there are numerous places where the language does not work or is confusing. If English is not the first language and the authors need assistance, I encourage them to reach to the journal editor.

Answer:

The text was proof-read to improve the grammar and the flow.

  1. General:

B.1.

References needed for sentences in lines: 61, 64-65, 88-89, 91-92, 93-94, 296-297, 354-355.

Answer:

The references were added to the manuscript.

B.2.

In the results and discussion, the authors switch between past and present tense! Please pick one and standardize throughout.

Answer:

The conjugation time was standardized in the results section and the discussion section.

B.3.

Whenever percentages or proportions are reported (for behavioral data and microbial data), if they are averages across multiple individuals, they need to also include plus/ minus standard deviations (same with figure 1, should include error bars).

Answer:

The average proportions and standard deviation of behavioral data and microbial data were added in the text of the new version of the manuscript. We also add error bar to figure 1.

B.4.

Figure are very small and difficult to read. This could be an artifact of initial submission but for final submission, the figures should be much larger and higher resolution.

Answer:

We have reintroduced the figures with a better quality.

B.5.

In discussion, there are places where it is unclear whether the results being discussed are from this study or previous study (e g  352-353 )  Be sure to clarify.

Answer:

The text in the manuscript has been reworked to make it clearer (Lines 404- 407).

  1. Specific

C.1.

72-75: There is a lot of debate about proportion of bacterial to hosts cells/ DNA, so be careful when reporting them! Here, they are not necessary for this study and I would remove them (see comment above about introduction).

Answer:

This information was removed.

C.2.

166-167: were sequences from chloroplast and mitochondria filtered out? Not always necessary but should be reported either way.

Answer:

Yes, they are.

C.3.

169: What were the min and the max sampling depths for the samples? Need to include a rarefaction plot at a supplementary figure to determine if 10,000 is an appropriate rarefying depth.

Answer:

A Goods coverage test is included to assess whether the depth of analysis is sufficient.

C.4.

192: Please define ‘sweet food’ It is very ambiguous and could include fruit.

Answer:

We define as Sweet foods the following items: cakes, wafers, biscuits, chocolate and sweets. This definition was added in the manuscript (Lines 228- 229).

C.5.

362-363: Akkermansia is no longer a monotypic genus.

Answer:

In fact, since 2016 there is a second species in the genus akkermansia, whose phylogenomic position is confirmed since 2019: 1 genus, 2 species: A. muciniphila and A. glycaniphila.  The sentence has been rectified in the manuscript (Line 413).

C.6.

Figure 1: The percentage add up to more than 100, please revise! Also, percentages is spelled incorrectly on the x-axis title.

Answer:

We have revised the figure 1 and make the corrections.

Round 2

Reviewer 2 Report

The authors have responded to the previous comments and the manuscript is much improved! There are a few comments to address before publication, but they are very minor.

Lines 88-89: Although the study of primate GMBs is limited compared to human GMBs, many would argue that primate GMBs have received more scientific attention than most other animals. The authors have done a great job revamping the introduction but I would change this sentence to make it clear that there is now a substantial body of literature on primate GMBs (on the order of thousands of published papers) that covers every family of primates.

Line 95: “host-related factors” is quite vague and would also apply to phylogeny. I would change to something that reflects how those traits can change over time e.g., “transient host-related factors”.

Line 101: change “funding” to “finding”.

Line 108: change “auteurs” to “authors”.

Line 126: please define GIT.

Figures:

Fig 2. Wow! That a lot of Firmicutes. This is a somewhat unexpected results compared to the GMBs of other primates. It might be worth noting in the discussion.

Figs 5 & 6: I would combine these figures into one so that the reader can easily compare the two metrics of beta diversity. I would make sure that the axes in the two figures are oriented the same way.

Author Response

Dear reviewer,

thank you for your insightful comments and corrections, we took the liberty to copy your comments below to answer here. The corrections made in the text were all applied as requested and have been marked in the article text.

We would like to thank you for the time spent reviewing our article.

Best regards,

For the Authors,

Dr. Moula

Question 1:

The authors have responded to the previous comments and the manuscript is much improved! There are a few comments to address before publication, but they are very minor.

Lines 88-89: Although the study of primate GMBs is limited compared to human GMBs, many would argue that primate GMBs have received more scientific attention than most other animals. The authors have done a great job revamping the introduction but I would change this sentence to make it clear that there is now a substantial body of literature on primate GMBs (on the order of thousands of published papers) that covers every family of primates.

Response:

The sentences have been modified (Lines 86- 87).

Question 2:

Line 95: “host-related factors” is quite vague and would also apply to phylogeny. I would change to something that reflects how those traits can change over time e.g., “transient host-related factors”.

Response:

“host-related factors” was replaced with “transient host-related factors”.

Question 3:

Line 101: change “funding” to “finding”.

Response:

it's done.

Question 4:

Line 108: change “auteurs” to “authors”.

Response:

it's done.

Question 5:

Line 126: please define GIT.

Response:

it's done.

Question 6:

Fig 2. Wow! That a lot of Firmicutes. This is a somewhat unexpected results compared to the GMBs of other primates. It might be worth noting in the discussion.

Response:

In fact, the proportion of Firmicutes found in the present study is higher than in other studies in primates.

Two additional sentences have been added to the discussion (Lines 366- 369).

Question 7:

Figs 5 & 6: I would combine these figures into one so that the reader can easily compare the two metrics of beta diversity. I would make sure that the axes in the two figures are oriented the same way.

Response:

it's done.
